# Beyond Static Cold Storage: Toward the Next Generation of Tailored Organ Preservation Solutions

**DOI:** 10.3390/ijms26199515

**Published:** 2025-09-29

**Authors:** Fernanda W. Fernandes, Fatma Selin Yildirim, Hiroshi Horie, Omer F. Karakaya, Chunbao Jiao, Geofia S. Crasta, Nasim Eshraghi, Koki Takase, Tobias Diwan, Laura Batista de Oliveira, Charles Miller, Chase J. Wehrle, Sangeeta Satish, Keyue Sun, Naoto Matsuno, Andrea Schlegel

**Affiliations:** 1Department of Inflammation and Immunity, Lerner Research Institute, Cleveland, OH 44195, USA; fernanf5@ccf.org (F.W.F.); jiaoc@ccf.org (C.J.); diwant2@ccf.org (T.D.);; 2Transplantation Center, Digestive Disease and Surgery Institute, Cleveland Clinic, OH 44195, USA; 3Department of Transplantation Technology and Therapeutic Development, Asahikawa Medical University, Asahikawa 078-8802, Hokkaido, Japan; 4Department of Clinical Engineering, National Center for Child Health and Development, Setagaya City 157-8535, Tokyo, Japan

**Keywords:** preservation solution, cold storage, machine perfusion, transplantation, outcomes

## Abstract

Machine perfusion technologies have redefined the landscape of organ preservation by enabling not just static cold storage, but graft optimization and assessment with the opportunity for additional therapeutic interventions. Preservation solutions, traditionally developed for static cold storage, are now being adapted for use in dynamic perfusion platforms. The optimal composition for machine perfusion remains unclear as we shift to creating biologically intelligent platforms tailored to mitigate ischemia–reperfusion injury. This review presents a mechanistic framework for understanding organ preservation through the lens of shared vulnerabilities, particularly: mitochondrial dysfunction, endothelial barrier breakdown, and the activation of inflammatory cascades. We discuss the evolution of classical preservation solutions, the rationale for redox-targeted and endothelial-stabilizing additives, and the promise of modular approaches adaptable to both static cold storage and machine perfusion. By integrating recent preclinical insights, systems biology, and emerging clinical trials, we outline the path toward unified, precision-preservation strategies capable of expanding the donor pool and improving transplant outcomes.

## 1. Introduction

Organ preservation is a cornerstone of successful transplantation, serving to maintain the structural and functional integrity of donor organs from procurement to implantation. Static cold storage (SCS) has enabled the widespread clinical adoption of transplantation by providing a simple, cost-effective, and logistically feasible method of slowing metabolic activity and minimizing ischemic injury [1,2]. Despite its strengths, the use of SCS alone is increasingly limited in the context of today’s transplant landscape.

Rising demand for organs has led to broader acceptance of organs from extended criteria donors (ECDs), which are grafts with increased risk factors, for example, advanced age, macrosteatosis (liver), increased creatinine (kidney), or collection after circulatory death (DCD). Organs from ECDs experience worse post-transplant outcomes and are more vulnerable to ischemia–reperfusion injury (IRI) with SCS alone [3,4,5]. To address this, the field is moving toward more sophisticated strategies aimed at improving organ viability and expanding the donor pool.

One key area of innovation is the use of machine perfusion, including hypothermic, subnormothermic, and normothermic modalities. This enables graft assessment, oxygen delivery, and targeted therapeutic intervention [6,7]. Unlike SCS, where solutions passively fill vessels to prevent collapse, machine perfusion applies pressure to actively circulate perfusate through the organ. This transforms the graft environment, introducing shear stress and recirculating metabolites. These effects are not yet fully understood. While these platforms offer powerful opportunities for intervention, they also present new physiological challenges. Optimizing perfusate composition may help mitigate oxidative stress, stabilize the endothelium, and support metabolic recovery before transplantation [1,2,8]. While many reviews describe the contents of standard preservation solutions, few detail how specific components act at key points in the IRI cascade. Here, we present a systems-based framework linking injury mechanisms to protective strategies and outline emerging directions for solution design.

## 2. Historical Evolution of Preservation Solutions

Ramos et al. [2] provide a comprehensive overview of the historical development and comparative studies of organ preservation solutions. Figure 1 illustrates a timeline of their clinical adoption for SCS, which for decades remained the cornerstone of transplantation practice. While EuroCollins emerged as the predominant solution in the 1960s, designed to approximate intracellular conditions and limit ion shifts during cold ischemia, important limitations became evident: rapid ATP depletion impaired energy-dependent ion transport (e.g., Na^+^/K^+^-ATPase), producing intracellular Na^+^/Ca^2+^ loading, osmotically driven water influx with cellular swelling, and compromised endothelial/glycocalyx integrity. The formulation’s lack of impermeants and robust oncotic support offered little counterforce to these fluid shifts. These processes contributed to early allograft dysfunction and inferior outcomes after transplantation [2,3,5].

A major advance came with the development of the University of Wisconsin (UW) solution, which addressed these issues through several innovations: inclusion of hydroxyethyl starch (HES) to prevent interstitial edema, addition of glutathione and allopurinol to mitigate oxidative stress, and impermeants such as lactobionate and raffinose to maintain osmolarity and ion balance [2,8]. These features made UW the gold standard for cold storage preservation, especially in liver and pancreas transplantation, and it remains the most widely used solution globally.

In contrast, histidine–tryptophan–ketoglutarate (HTK) was developed as a low-viscosity alternative to permit single-dose, high-volume organ perfusion. Its composition focuses on membrane stabilization, using histidine for buffering, tryptophan for membrane protection, and alpha ketoglutarate to support intermediary metabolism [2]. Because HTK contains no macromolecular oncotic agent, it provides limited counter-oncotic support during cold ischemia. The risk of edema therefore reflects the ischemic context and the total flush volume rather than the solution alone. Historically, higher volumes have been recommended for HTK (e.g., 8–15 L), whereas UW is often used in lower volumes; however, contemporary procurement practice frequently uses HTK and UW at equivalent volumes, which preserves HTK’s potential per-liter cost advantage, yielding per-donor savings in some procurement settings. Thus, the net cost-effectiveness of HTK is protocol-dependent: higher volumes can offset price differences, whereas equivalent-volume HTK regimens retain the economic benefit [9,10].
Figure 1Evolution of organ preservation solutions. This timeline chart illustrates the development and predominant clinical use of preservation solutions across static cold storage (SCS), hypothermic machine perfusion (HMP/HOPE), and normothermic machine perfusion (NMP), including ex vivo lung perfusion (EVLP). Solutions are categorized by platform with icons denoting primary organ application. HOC now mostly used for flushing, not cold storage (gradient phasing out). Normothermic perfusates generally consist of packed red blood cells with additives (e.g., electrolytes, glucose, amino acids, heparin, oncotic agents like albumin or Gelofusine), whereas STEEN™ is the only routinely used acellular perfusate for EVLP [1,2,11,12,13,14,15,16]. HOC: Hyperosmolar Citrate; UW: University of Wisconsin; HTK: Histidine-Tryptophan Ketoglutarate; IGL: Institut Georges Lopez; IRI: ischemia–reperfusion injury; PEG-35: polyethylene glycol 35; ATP: adenosine triphosphate; HES: hydroxyethyl starch. * Raffinose and lactobionic acid as impermeants. HES as colloid. Allopurinol to reduce oxidative stress. Glutathione as free-radical scavenger. Adenosine as ATP substrate. ** Higher K^+^ to induce diastolic arrest. Mannitol as impermeant. Histidine as buffer, tryptophan as antioxidant. No colloid; less viscous than UW.
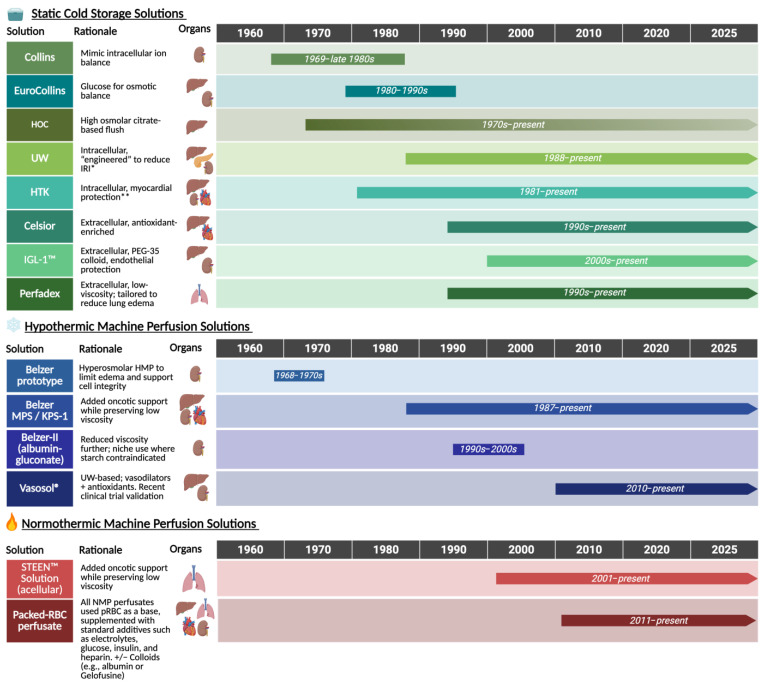


Building on these predecessors, Institut Georges Lopez (IGL-1) solution introduced further refinements by replacing HES with polyethylene glycol (PEG), which maintains endothelial barrier function with lower viscosity [17]. IGL-2 also adds adenosine, glutathione, and sodium nitrite to support mitochondrial respiration and counteract oxidative injury [18,19]. These compositional shifts reflect a broader understanding of the molecular mechanisms that drive ischemic injury and lay the foundation for therapeutic perfusates capable of organ protection and repair. Other agents, such as trehalose, have been explored for their membrane-stabilizing and osmo-protective effects [20].

Parallel to these biochemical advances, contemporary practice follows a simple sequence: after cross-clamp the graft is flushed intravascularly with cold preservation solution to wash out blood and induce rapid hypothermia. It is then either maintained in static cold storage (SCS), where buffers, impermeants, and oncotic agents help limit acidosis and edema, or connected to a machine-perfusion circuit. Hypothermic oxygenated perfusion (HOPE) supplies dissolved oxygen at low temperature, whereas normothermic machine perfusion (NMP) maintains near-physiologic metabolism; both approaches permit collection of intra-perfusion readouts to guide practice, including hemodynamic parameters (flows, pressures, vascular resistance), oxygen consumption/extraction, perfusate chemistry and acid–base status, and biochemical markers such as FMN [6,21,22]. While not yet globally adopted, these approaches are particularly promising for ECD grafts that may not tolerate static storage well.

As reliance on ECDs grows together with a general understanding of the role of an individual metabolic profile, static cold storage alone often fails to protect against complex IRIs. This has driven a shift toward more mechanistic strategies targeting shared vulnerabilities, particularly mitochondrial dysfunction, endothelial injury, and inflammation. The next section presents a systems-based framework addressing these injuries through preservation solutions and their pharmacological modulation.

## 3. Mechanisms of Injury and Protection: Systems-Based Framework

Organ preservation solutions have traditionally aimed to prevent cellular swelling and maintain electrolyte balance during cold storage. Intracellular-type solutions such as UW and IGL-1 use high potassium and low sodium to reduce ion shifts when ATP-dependent pumps slow; strong impermeants (lactobionate, raffinose) limit cellular water entry; buffers and antioxidants stabilize pH and redox state; an oncotic agent (HES in UW, PEG-35 in IGL-1) counteracts capillary filtration and supports the endothelium. Extracellular-type solutions such as HTK/Custodiol, Celsior for heart, and Perfadex for lung favor higher sodium and lower potassium to keep viscosity low for uniform flushing; HTK relies on histidine buffering with tryptophan and α-ketoglutarate and contains no macromolecular colloid; Perfadex includes dextran-40 to support the lung microcirculation. Machine-perfusion perfusates are formulated for continuous flow: Belzer MPS/KPS-1 is used widely for hypothermic perfusion of kidneys and, in some centers, livers; STEEN Solution supports ex vivo lung perfusion with human albumin and dextran-40 to provide colloid osmotic pressure. Figure 2 illustrates the current issues that occur during IRI, particularly oxidative stress, mitochondrial dysfunction, and endothelial injury, and how organ preservation solution components act to address these. This section outlines the rationale for key additives in modern preservation solutions, emphasizing their biologic mechanisms and the supporting evidence base. While many of these interventions are conceptualized for storage, their effect may be more targeted during machine perfusion, which allows for dynamic, targeted treatment.

### 3.1. Mitochondrial Dysfunction and Redox Imbalance

Reactive oxygen species (ROS) are key mediators of IRI, particularly during the reperfusion phase when the abrupt reintroduction of oxygen under normothermic temperatures meets a build-up of reduced electron carriers in the mitochondria due to impaired electron transport during prior ischemia [23,24,25,26]. Among the earliest mechanistic insights into this process was the identification of mitochondrial complex I as a dominant source of ROS following ischemia. Chouchani et al. [23] demonstrated that succinate accumulation during ischemia fuels reverse electron transfer (RET) through complex I upon reperfusion, generating a burst of superoxide at the flavin mononucleotide (FMN) site. This burst is especially injurious due to its proximity to mitochondrial membranes, where ROS trigger lipid peroxidation and disrupt membrane integrity. Inhibition of RET, either directly at complex I with rotenone or indirectly inhibiting complex II with dimethyl malonate, markedly reduced tissue injury in murine models [23,27]. While complex I is the dominant source of early ROS, complex II (succinate dehydrogenase (SDH)), plays an upstream role by generating the succinate pool during ischemia. Therefore, transient inhibition of complex II during cold storage by limiting succinate accumulation, with preclinical models suggesting that such inhibition is reversible and does not impair post-reperfusion mitochondrial function [28,29].

The ROS generated at the complex I FMN-binding site also provides a biomarker of IRI. Structural destabilization at this site leads to mitochondrial complex I FMN release, which can be measured spectroscopically from various transplant fluids in real-time [22]. Elevated FMN levels measured during liver and kidney perfusion have been shown to predict poor graft outcomes [22,30,31]. Markers such as malondialdehyde (MDA), indicating lipid peroxidation, and TNF-α (a cytokine released from immune and parenchymal cells during early reperfusion) are used to assess oxidative damage and inflammatory activation [30,31,32,33]. Figure 3 illustrates mitochondrial ischemia and reperfusion mechanisms, alongside protective strategies targeting complex I and II modulation, ROS scavenging, and control of membrane potential. Following Chouchani’s framework [23], these interventions converge on the succinate–RET axis as a central driver of reperfusion injury. Although preventing succinate accumulation during ischemia would be ideal, this is rarely feasible due to limitations in treating donors prior to procurement. Moreover, although malonate is effective at blocking succinate oxidation, it is irreversible and may impair mitochondrial recovery if retained in the graft at reperfusion. Consequently, strategies have shifted toward modulating mitochondrial activity during the reperfusion phase, where the risk of injury is greatest. Controlled oxygen reintroduction during HOPE slows succinate oxidation, preventing the abrupt ROS surge typically seen with normothermic reperfusion.

#### 3.1.1. Glutathione

To buffer the surge in reactive oxygen species (ROS), preservation solutions often include antioxidants. Glutathione (GSH) is the predominant intracellular antioxidant under physiologic conditions and plays a central role in neutralizing peroxides generated during oxidative stress. During cold ischemia, tissue GSH levels fall significantly, impairing redox homeostasis in renal, cardiac and hepatic models [34]. GSH functions as a cofactor for glutathione peroxidase (GPx), which detoxifies hydrogen peroxide (H_2_O_2_) and lipid hydroperoxides by reducing them to water and alcohols. This step is key because H_2_O_2_, while more stable than other radicals, can persist and contribute to further ROS generation and ongoing inflammation and cell death. During this enzymatic reaction, GSH is oxidized to glutathione disulfide (GSSG) and must be recycled to GSH by glutathione reductase in an NADPH-dependent reaction [25]. Disruption of this cycle due to GSH depletion, reduced NADPH availability or enzyme inhibition leads to H_2_O_2_ accumulation. In the presence of free iron, H_2_O_2_ can produce highly reactive hydroxyl radicals. These radicals cause oxidative damage to mitochondrial membranes, proteins, and mitochondrial DNA (mtDNA). This injury undermines mitochondrial function, impairs cellular viability, and contributes to graft dysfunction following transplantation [26]. GSH is a standard component of UW preservation solution (the gold standard for static cold storage), added at approximately 3 mM to neutralize ROS generated during cold ischemia and reperfusion [2]. However, GSH oxidizes during storage, and its effective concentration may be substantially lower if not freshly added at the point of use [35]. A randomized study comparing UW with vs. without supplemental GSH in both cold storage and machine perfusion found no significant differences in early renal graft outcomes; indeed, the use of machine perfusion alone had a greater impact on immediate function and delayed graft function than GSH supplementation [36]. Furthermore, GSH supplementation of UW has been reported to exacerbate endothelial dysfunction in cold-stored vascular tissue, raising concerns about its stability and reactivity in static conditions [37]. However, emerging data from machine perfusion suggest that GSH may still confer benefits under dynamic perfusion settings. In a randomized controlled trial of hypothermic oxygenated perfusion (HOPE) for liver transplantation, van Rijn et al. (2021) reported a significant reduction in non-anastomotic biliary strictures when using a perfusate that included GSH, highlighting its potential utility when oxygen and flow conditions enable active ROS detoxification mechanisms [38]. Nonetheless, GSH has limitations as a direct supplement: it is unstable in aqueous solutions, particularly at alkaline pH or in the presence of metals and is poorly taken up by cells due to its polarity. To overcome these limitations, alternative strategies have focused on the use of *N*-acetylcysteine (NAC), a stable, cell-permeable cysteine donor that enhances intracellular GSH synthesis. In preclinical kidney perfusion models, NAC preserved GSH levels, reduced oxidative stress, and improved renal histology [39]. Clinical data from liver transplantation also suggest NAC may improve early graft function, although these studies were not conducted under machine perfusion conditions [40].

#### 3.1.2. MitoQ

Additional antioxidant strategies include the use of targeted scavengers such as MitoQ, a mitochondria-targeted antioxidant consisting of a ubiquinone moiety attached to a lipophilic triphenylphosphonium (TPP^+^) cation. The TPP^+^ moiety enables MitoQ to selectively accumulate within the mitochondrial matrix, driven by the organelle’s negative membrane potential. Once inside, the ubiquinone portion of MitoQ is reduced to ubiquinol, which directly scavenges mitochondrial ROS such as superoxide and hydrogen peroxide, thereby preventing oxidative damage to mitochondrial lipids, enzymes, and mtDNA. This targeted delivery enhances efficacy while minimizing systemic effects. Preclinical studies support MitoQ’s ability to reduce mitochondrial injury during ischemia–reperfusion injury (IRI). In a rat model of kidney donation after circulatory death, MitoQ administered during hypothermic machine perfusion improved renal function, preserved ATP content, reduced mitochondrial swelling, and limited tubular injury [41]. Similarly, in a porcine kidney cold preservation model, MitoQ blunted mitochondrial damage and reduced injury during static cold storage [42]. In vivo murine studies also demonstrate protective effects in both kidney [43] and heart transplantation models [44], where MitoQ preserved mitochondrial ultrastructure and function, reduced oxidative biomarkers, and improved graft viability. A recent review further highlights mitochondrial dysfunction as a central driver of kidney injury, emphasizing the therapeutic relevance of targeted antioxidants like MitoQ in renal IRI [45]. Collectively, these findings propose MitoQ as a proof-of-concept compound for redox-targeted mitochondrial protection in organ preservation protocols.

#### 3.1.3. Itaconate

Itaconate emerged from inflammation studies, where it is produced by activated macrophages via decarboxylation of cis-aconitate through immune-responsive gene 1 (IRG1). It was first identified as an endogenous inhibitor of mitochondrial SDH limiting succinate oxidation and thus blunting ROS generation during reperfusion [29]. Since cold storage involves suppressed metabolism, transient SDH inhibition is well tolerated and reversible upon reperfusion. Itaconate also activates the Nrf2 antioxidant pathway via alkylation of KEAP1 cysteine residues, thereby lifting repression of cytoprotective gene expression [46]. Together, these mechanisms confer a dual protective role: direct inhibition of ROS at their mitochondrial source and transcriptional upregulation of endogenous defenses. Due to limited membrane permeability of native itaconate, more bioavailable derivatives such as dimethyl itaconate (DMI) and 4-octyl itaconate (4-OI) have been developed and tested in preclinical IRI models across multiple organ systems. In murine hepatic IRI, 4-OI reduced histological liver damage and inflammatory cytokine levels, with transcriptomic analyses highlighting involvement of lncRNA–mRNA regulatory networks [47]. In acute kidney injury models, 4-OI activated Nrf2 while suppressing STAT3 signaling to mitigate injury [48]. In models of sepsis-induced systemic inflammation, 4-OI preserved mitochondrial integrity, reduced oxidative damage, and regulated immune responses via Nrf2–PD-L1 modulation [49]. Finally, in myocardial reperfusion models, 4-OI promoted angiogenesis and limited endothelial hypoxia through ERK pathway activation [50]. While direct application of these compounds in organ preservation systems such as HOPE has yet to be clinically validated, their mechanistic breadth and organ-specific efficacy support further translational exploration, particularly for enhancing redox and endothelial stability in marginal grafts.

#### 3.1.4. Quercetin

Quercetin is a dietary flavonoid with potent antioxidant and anti-inflammatory properties. It scavenges ROS, inhibits NF-κB, and activates mitochondrial biogenesis pathways such as PGC-1α and SIRT1, supporting mitochondrial recovery after IRI [51,52]. Mechanistically, it also stabilizes mitochondrial membranes and mitigates oxidative injury by modulating intracellular calcium and redox signaling [53]. In preclinical models of ex situ liver preservation, quercetin reduced lipid peroxidation, preserved mitochondrial architecture, improved bile production, and maintained endothelial morphology during perfusion [54]. Similarly, in porcine kidney transplantation, quercetin–sucrose preservation solutions improved graft histology, reduced tubular injury, and preserved both renal function and endothelial integrity after cold ischemia [55,56]. However, its clinical translation is challenged by poor solubility and rapid hepatic phase II metabolism, including glucuronidation and sulfation, which reduce bioavailability during reperfusion [57]. In the context of ex situ organ perfusion, direct endothelial exposure to quercetin may help circumvent these limitations, supporting continued investigation into its use as an endothelial- and mitochondria-targeted additive in machine perfusion systems.

#### 3.1.5. Hydrogen Sulfide Donors

Hydrogen sulfide (H_2_S) has emerged as a promising adjunct in organ preservation due to its ability to reduce IRI. Insights from hibernating species, which naturally endure low metabolic states without organ damage, suggest that H_2_S can mimic this protective effect by suppressing mitochondrial activity and limiting oxidative stress during rewarming [58]. Preclinical studies using mitochondria-targeted H_2_S donors such as AP39 have shown protective effects in both myocardial and renal ischemia models by ROS, inhibiting mitochondrial permeability transition, and preserving graft architecture and function [59,60]. Dugbartey et al. [60] demonstrated that supplementing UW solution with AP39 during static cold storage significantly improved renal function, oxygenation, and histological outcomes in a porcine DCD kidney transplant model. Broader studies of H_2_S in various IRI models have confirmed its antioxidant, anti-inflammatory, and anti-apoptotic actions [61]. These findings highlight the therapeutic potential of incorporating H_2_S donors into preservation strategies to protect grafts at the mitochondrial level, though their role in machine perfusion remains to be defined.

#### 3.1.6. PrC-210

PrC-210 is a thiol-based, non-enzymatic reactive oxygen species (ROS) scavenger designed to neutralize hydroxyl radicals and superoxide anions immediately upon exposure. Unlike enzymatic antioxidants, PrC-210 diffuses rapidly into tissues and directly quenches ROS at the site of generation, making it a compelling candidate for reducing oxidative injury during organ preservation. In a rat kidney transplant model using UW solution supplemented with 30 mM PrC-210 during cold storage, histologic damage and markers of apoptosis (e.g., activated caspase-3) were significantly reduced, alongside improvements in creatinine, BUN, and inflammatory cytokines [62,63]. Additional studies have shown that even brief exposure (15 s) to PrC-210 in UW solution before cold ischemia markedly reduced ROS-mediated DNA damage, preserved brush border integrity, and suppressed renal tubular necrosis for up to 30 h of storage [63]. Furthermore, systemic pre-treatment with PrC-210 prior to ischemia in murine models significantly attenuated renal tubular injury and decreased serum BUN and caspase activity [64]. These results collectively support the role of PrC-210 as a potent, immediate-acting ROS buffer capable of protecting organ integrity during static cold storage. However, its performance during machine perfusion remains uncharacterized and warrants investigation.

### 3.2. Endothelial Barrier Breakdown and Vascular Dysfunction

The endothelium serves as both a physical barrier and a dynamic interface regulating vascular tone, immune surveillance, and permeability. Preservation injury affects not only the parenchyma but also this critical vascular layer, impairing endothelial junctional integrity and triggering increased permeability, leukocyte adhesion, and microcirculatory failure (Figure 1), a shared injury pathway in many graft types [65,66].

These changes have been observed after SCS in multiple vascular beds, including cardiac and hepatic allografts, with histological findings of endothelial swelling, vacuolization, and detachment [67,68,69,70,71]. Tight junctions, adherens and the glycocalyx are among structures shown to be particularly susceptible to hypothermia, oxidative stress, and proteolytic degradation [67,68,70,71,72].

Sinusoidal endothelial cells (LSECs) are particularly vulnerable to ischemia and oxidative injury, with cold-induced ferroptosis emerging as a key mechanism of damage linked to poor graft function [73]. Yet this susceptibility is not confined to the liver: across renal, cardiac, and pulmonary grafts, ischemia triggers endothelial activation, ICAM-1 upregulation, and barrier breakdown [74,75].

These effects reflect a broader metabolic fragility of the endothelium. Recent studies emphasize mitochondrial dysfunction as a central driver of endothelial injury: disrupting redox balance, nitric oxide production, and flow-mediated mechanotransduction [75]. While LSECs provide a model of this axis, mitochondrial-endothelial coupling likely represents a conserved vulnerability across organs and preservation platforms.

Preservation solutions attempt to mitigate endothelial dysfunction via osmotic support, antioxidant delivery, and metabolic modulation. Colloids are foundational to this approach: hydroxyethyl starch (HES), used in UW solution, and polyethylene glycol-35 (PEG-35), used in IGL-1/2, exert oncotic pressure to limit interstitial edema. PEG-35 outperforms HES in maintaining endothelial ultrastructure and glycocalyx integrity during hypothermic conditions, with superior outcomes demonstrated in both liver and kidney models [2,8].

Similarly, nitric oxide (NO) donors stabilize sinusoidal perfusion and downregulate endothelial activation markers like ICAM-1 and eNOS in hypothermic perfusion models [76]. Flavonoids like quercetin enhance endothelial resilience through antioxidant and anti-inflammatory pathways. In kidney SCS, quercetin reduced NF-κB and ICAM-1, upregulated ERG (a transcription factor key to endothelial stability), and decreased platelet adhesion [55]. Exogenous corticosteroids have also attenuated endothelial activation in liver normothermic machine perfusion (NMP) by reducing ICAM-1 levels [69].

The beneficial effects of endothelial protection are further supported by dynamic perfusion studies, where reduced ROS and lactate production often correlate with preserved microvascular tone [32]. In addition, ALDH2-mediated modulation of oxygen reserve capacity has been linked to endothelial senescence under ischemic stress [18,77], reinforcing the concept that metabolic support is integral to vascular integrity.

Together, these findings underscore that endothelial dysfunction is a universal graft vulnerability, not an organ-specific phenomenon. Table 1 summarizes endothelial-specific findings with different preservation solutions. A therapeutic strategy focused on preserving endothelial structure and mitochondrial health supports the rationale for unified preservation platforms. This is exemplified by the success of IGL-2 and other endothelial-targeted solutions across multiple organs and perfusion modalities [2]. Integrating redox modulation with structural protection may simplify preservation logistics while expanding graft viability and transplant access.

### 3.3. Inflammatory Signaling and Immune Activation

While endothelial dysfunction is an early trigger of IRI, it initiates a broader cascade of immune activation that extends beyond the vasculature. The inflammatory response not only amplifies vascular injury but also drives leukocyte infiltration, cytokine release, and long-term graft immunogenicity. As such, preservation strategies are increasingly designed to modulate not just structural injury, but also the immune landscape of the graft.

#### 3.3.1. Nitric Oxide and Immune Modulation

Nitric oxide (NO) donors, such as nitroglycerin and S-nitroso-N-acetylpenicillamine, reduce leukocyte adhesion, suppress endothelial activation, and promote vasodilation through cyclic GMP signaling. These effects improve microvascular perfusion and limit early immune cell recruitment during liver and kidney preservation. In HOPE models, NO supplementation has been shown to stabilize sinusoidal flow and reduce expression of adhesion molecules such as ICAM-1 [76]. Recent studies further underscore the role of endogenous NO synthesis in graft protection. Restoration of NOS1β expression in the macula densa through bicarbonate-induced alkalization significantly improved renal graft outcomes in a mouse model by enhancing NO bioavailability [79]. These findings reinforce NO’s dual role in vascular and immune regulation during organ preservation.

#### 3.3.2. Prostaglandins and Cyclooxygenase Inhibitors

Prostaglandin E1 (PGE1) exerts potent vasodilatory and anti-platelet effects, improving perfusion and supporting tissue oxygenation during preservation. Beyond these hemodynamic actions, PGE1 modulates innate immune activation by suppressing damage-associated molecular pattern (DAMP) signaling pathways—reducing expression of TLR4, NF-κB, and downstream pro-inflammatory cytokines such as IL-6 and TNF-α [80]. These effects have been validated in both kidney and liver models, where PGE1 administration before or during cold preservation attenuated histological injury and improved post-transplant function [80,81]. Mechanistically, PGE1 limits neutrophil accumulation, stabilizes mitochondrial membrane potential, and promotes antioxidant gene expression. It is also a key component of Vasosol™ [2], a proprietary oxygenated perfusate designed for arterial infusion during hypothermic perfusion, highlighting its clinical relevance as both an anti-inflammatory and perfusion-enhancing agent.

#### 3.3.3. Immune Modulation via Perfusion Platforms

Machine perfusion itself may influence the immune environment of the graft. Ex situ liver perfusion has been shown to downregulate pro-inflammatory pathways while promoting genes involved in tissue repair and immune tolerance [76]. Transcriptomic analysis of liver grafts post-HOPE revealed suppression of IL-6, TNF-α, and HIF-1α, as well as reduced recruitment of neutrophils and T cells. These immune effects may stem from improved endothelial integrity and metabolic stabilization but also reflect perfusion-mediated modulation of innate immune sensors such as Toll-like receptors (TLRs) and NOD-like receptors (NLRs).

### 3.4. Energy Collapse and Metabolic Support

Mitochondrial metabolism is severely impaired during ischemia and early reperfusion, leading to ATP depletion, ROS production, and downstream cell injury. To counter this, preservation solutions have incorporated metabolic substrates that support mitochondrial recovery and energy generation. Adenosine, a purine nucleoside included in Belzer MPS, facilitates ATP resynthesis and modulates purinergic signaling. Its inclusion has been shown to preserve mitochondrial respiration, enhance post-reperfusion ATP levels, and improve cellular viability in renal and hepatic models [2,3]. Similarly, amino acids such as glutamate and aspartate, originally added to solutions like HTK and Bretschneider’s, replenish tricarboxylic acid (TCA) cycle intermediates, supporting mitochondrial integrity and reducing oxidative stress during cold storage and reperfusion [1,4]. These substrates contribute to energy homeostasis and may synergize with redox stabilizers to preserve mitochondrial function. While preclinical studies underscore the benefit of these additives, their long-term impact on metabolic signaling pathways and interaction with other perfusate components remain areas for further investigation. Overall, targeted substrate delivery offers a rational strategy to enhance mitochondrial resilience during organ preservation.

### 3.5. Pharmacologic Modulators and Emerging Additives

Alongside standard preservation components, pharmacologic agents are being actively investigated for their potential to modulate endothelial signaling and blunt ischemia-induced injury pathways. Beta-blockers, including carvedilol and propranolol, have shown protective effects in reperfusion models by inhibiting adrenergic-driven oxidative stress and limiting calcium overload. In isolated heart and liver systems, propranolol has been associated with reduced lipid peroxidation and improved post-ischemic recovery. These mechanisms are potentially mediated via mitochondrial and membrane stabilization [4]. Tacrolimus, traditionally used as an immunosuppressant, also exerts direct cytoprotective effects on endothelial and parenchymal cells. These effects appear independent of T-cell modulation and involve preservation of mitochondrial membrane potential, inhibition of cytochrome c release, and suppression of caspase-mediated apoptosis during cold ischemia [4]. Modulators of inflammatory signaling, such as AG490, a JAK2/STAT3 inhibitor, have also demonstrated benefit in preclinical cold ischemia models by reducing pro-inflammatory cytokine expression and dampening apoptosis in renal grafts [4]. Further supporting these pharmacologic approaches, a 2024 study by Bejaoui et al. [33] evaluated PERLA^®^, a novel PEG- and gluconate-based cold storage solution supplemented with antioxidant and anti-inflammatory agents. In a rat kidney transplantation model, PERLA^®^ significantly reduced oxidative injury and improved early graft function, highlighting the translational potential of integrated pharmacologic formulations. Together, these agents offer a mechanistically diverse toolkit for modulating IRI and enhancing graft viability. Their selective use within perfusion systems or static storage may help extend preservation timeframes.

## 4. Organ-Specific Preservation and the Case for Unification

### 4.1. Variability in Injury Profiles Across Organs

Each organ exhibits distinct physiological and metabolic characteristics that influence its susceptibility to IRI and determine its preservation requirements. For instance, the lungs are directly exposed to oxygen and more tolerant of oxidative and thermal stress, making them amenable to preservation across a broader temperature range. In contrast, the liver has a dense mitochondrial network and high metabolic demand, rendering it particularly vulnerable to cold ischemia and RET; HOPE has therefore been shown to improve outcomes in liver transplantation [38,82], with broader perspectives on liver machine-perfusion modalities and endpoints reviewed in [83]. Kidneys are relatively ischemia-tolerant due to their low basal oxygen consumption and autoregulatory capacity but remain susceptible to tubular injury during prolonged cold storage [21]. The heart’s reliance on tightly regulated vascular flow and contractile function makes it sensitive to microvascular dysfunction and electrolyte imbalances during preservation. Uterine grafts (though still experimental) also present unique challenges such as hormonal sensitivity and limited data to guide practice [84,85,86]. These differences underscore that no single strategy supports all grafts equally.

### 4.2. Limitations of the One-Size-Fits-All Approach

Despite well-documented differences in organ biology, current clinical practice still relies on a limited set of static preservation solutions, primarily UW and HTK, which were originally developed for liver and kidney storage [2,5]. These solutions are commonly used across a broad range of organs with minimal adaptation, even when their physicochemical properties may not align with the physiology of the graft. HTK’s low viscosity facilitates perfusion and flushing but lacks colloidal support for endothelial protection. In contrast, UW contains hydroxyethyl starch for oncotic balance but has a high potassium content and viscosity that may impair microvascular flow, particularly in cardiac and pulmonary grafts [2,67]. Importantly, these formulations are static in composition and do not accommodate variability in graft quality, ischemic exposure, or preservation platform. This generalized approach overlooks the biological and logistical heterogeneity of transplantation and may limit optimization for high-risk grafts.

### 4.3. Towards Unified or Modular Strategies

Evidence increasingly supports the rationale for a modular preservation framework, a strategy that combines a shared base solution targeting common injury mechanisms with the flexibility to incorporate additives tailored to organ type, ischemic risk, and preservation modality. Such a framework would enable preservation protocols to be adapted in real time based on measurable graft vulnerabilities and expected stressors. This model differs from current practice by allowing targeted interventions rather than relying on a single formulation for all grafts. For instance, grafts with prolonged warm ischemia or DCD status may benefit from enhanced antioxidant or mitochondrial support during machine perfusion, whereas lower-risk grafts might require only baseline stabilization during static cold storage. While not yet implemented widely in clinical transplantation, this approach reflects an evolving view that preservation strategies should be personalized and platform-specific, especially as dynamic perfusion technologies and biomarker-guided monitoring continue to advance.

### 4.4. Divergence of Storage and Perfusion Solutions

As preservation strategies shift from SCS to dynamic MP, it becomes increasingly important to distinguish between solutions designed for passive preservation and those intended for active circulation. While many preservation solutions are used interchangeably across platforms, their composition, viscosity, and pharmacodynamic roles differ significantly depending on the delivery context. Some solutions such as UW Machine Perfusion Solution (Belzer MPS), KPS-1, and IGL-1, are specifically formulated for both cold storage and hypothermic perfusion, offering low viscosity, oncotic support, and redox stability. These formulations are optimized to withstand circulation and minimize shear stress, while still preserving mitochondrial and endothelial integrity under cold conditions. By contrast, traditional storage solutions like UW (standard), HTK, and Celsior were developed for static preservation and are frequently repurposed for machine perfusion despite limited validation in that context. This practice introduces variability in clinical outcomes and complicates mechanistic interpretation, as these solutions were not originally designed for active circulation. Perfusate composition becomes even more variable under normothermic machine perfusion (NMP), where solutions must support full metabolic activity. NMP perfusates typically combine a crystalloid base (e.g., Ringer’s, albumin, or saline) with oxygen carriers—most commonly red blood cells (RBCs), although hemoglobin-based and acellular alternatives are also under investigation [87]. The additive landscape remains heterogeneous, ranging from vasodilators and bile salts to antioxidants and anti-inflammatory agents. Despite these advances, the fluid matrix of NMP remains underexplored and lacks standardization. This distinction is critical when interpreting the literature or designing future trials. These issues are synthesized for liver MP in [83]. For an overview of solutions used across platforms and their proposed applications, see Table 2A,B.

### 4.5. Temperature Shift in Cardiothoracic Preservation and Implications

The move toward normothermic perfusion in thoracic programs and the wider use of oxygenated hypothermia in abdominal grafts do more than swap one fluid for another. They require temperature-specific formulations (low-viscosity, oxygen-rich solutions that stabilize mitochondria and endothelium at hypothermia; albumin- or blood-based carriers with active buffering at normothermia) and they reshape workflows (circuits, monitoring, and staffing for warm runs versus simpler logistics for HOPE). Assessment tools also diverge by temperature: readouts collected during hypothermia and normothermia are not interchangeable and should be interpreted in context rather than as universal pass–fail tests. The economics differ as well: warm perfusion adds equipment and consumables, whereas hypothermic oxygenated strategies add time but can be implemented with simpler perfusates. Taken together, these trends argue against one formulation for all settings and reinforce the need for the modular model outlined above selecting a shared base and adding components by organ, injury profile, and target temperature, with endpoints and trial designs matched to the platform.

## 5. Preservation Modalities: Modulating Injury Through Storage and Perfusion

Machine perfusion technologies have rapidly evolved over the last few decades, now representing an active platform for organ protection, functional assessment, and targeted therapeutic intervention. Unlike SCS, which limits metabolic demand, machine perfusion enables continuous oxygen and nutrient delivery, active metabolism of toxic compounds, and real-time adjustment of perfusate composition tailored to each graft’s needs [6,66].

Multiple modalities have emerged, each with unique advantages. HOPE maintains lower metabolic demands while restarting mitochondrial respiration at slower rates compared to normothermic conditions, thereby rebuilding ATP and metabolizing succinate with limited ROS release during HOPE and later upon rewarming and transplantation. Subnormothermic and normothermic machine perfusion (NMP) further induce a higher metabolic and synthetic activity, enabling viability assessment and providing a therapeutic window for molecules to actively interfere with organ metabolism before transplantation. This higher temperature comes however at the price of more severe mitochondrial injury and IRI [5,32].

Machine perfusion also allows for phase-dependent delivery of therapeutic agents, enabling the administration of unstable or oxygen-sensitive compounds at optimal time points. For instance, hemoglobin-based oxygen carriers (HBOCs) and precise oxygen tension modulation may reduce mitochondrial anoxia and enhance ATP regeneration, while the continuous or pulsed infusion of additives, such as antioxidants (e.g., MitoQ, hydrogen sulfide donors), PEG-35, metabolic buffers, or corticosteroids, can promote endothelial and mitochondrial protection [1,4,8,21,24,32,41,42,43,44,97,98].

Moreover, machine perfusion facilitates real-time graft evaluation and enables adaptive perfusate modification in response to metabolic feedback, establishing a framework for truly personalized preservation. For instance, FMN measurement has undergone international methodological and multicenter validation in liver HOPE, with pilot data supporting its use in graft assessment; applications to kidney and pancreas remain under investigation [22,30,31]. The ability to track such dynamic injury markers not only informs transplant decisions but also opens a therapeutic window: marginal grafts initially deemed unsuitable may respond to targeted interventions (such as antioxidants or mitochondrial protectants) with measurable improvement in biomarkers like FMN, supporting their rescue and clinical use.

## 6. Summary and Future Outlooks

Organ preservation is undergoing a profound transformation, from a passive strategy designed to delay ischemic injury to a dynamic therapeutic opportunity. As this review has outlined, the shared mechanisms of graft damage, including mitochondrial redox imbalance, endothelial dysfunction, and inflammatory signaling, provide a compelling rationale for cross-organ, mechanism-based intervention [1,2,3,6,34]. The growing body of experimental and clinical work demonstrates that machine perfusion platforms, once limited to transport logistics, now enable real-time assessment, targeted therapy, and the modulation of graft response [11,22,31,82].

However, as preservation strategies evolve, so must their implementation. Different organs exhibit distinct physiological demands and vulnerabilities—necessitating more nuanced, context-specific approaches [2,8,28]. Cold storage will likely remain the global standard in many settings, yet the integration of perfusion technologies opens the door for personalized, phase-specific delivery of therapeutic compounds [4,21]. While universal solutions may offer logistical advantages, emerging evidence supports the development of modular preservation strategies tailored to the organ, condition, and anticipated risk [5,31].

The future lies at the interface of biotechnology, computational modeling, and systems biology. Multi-omics tools (e.g., transcriptomics, metabolomics, proteomics) are refining our mechanistic understanding of ischemia–reperfusion injury [8,23], while machine learning approaches are beginning to inform perfusion protocols, viability thresholds, and individualized interventions [46,99]. As shown in recent pilot studies, biomarkers like FMN can guide dynamic decision-making during perfusion [22,30,31], potentially rescuing marginal grafts that would otherwise be discarded [19,33].

Nonetheless, clinical translation remains constrained by significant barriers. Pathways to market vary across regions and product types, and this slows iteration. In the United States, most cold storage fluids and some hypothermic perfusion solutions are regulated as medical devices and are often cleared by showing similarity to an existing product; examples include UW/SPS-1, Custodiol HTK, KPS-1, and the recently cleared Belzer MPS for continuous hypothermic perfusion of explanted abdominal organs. Platforms that keep an organ functioning at warm temperature are reviewed as higher-risk technologies and generally require prospective clinical evidence before broad use; recent approvals include EVLP with XPS/STEEN and the OCS Liver system [100]. In Europe and the UK, classifications diverge: several preservation solutions are licensed as medicines with drug-style labeling and safety monitoring (for example, Custodiol HTK has a UK SmPC), while pumps and disposables are certified as medical devices under a separate rulebook with extended transition timelines. The result is higher cost and longer timelines, fewer opportunities for small but important reformulations, and a tendency for established solutions to remain dominant even as the science advances.

Beyond regulation, implementation is also limited by how organs are procured and evaluated. Procurement practices vary across centers, and endpoints are not standardized, which makes results hard to compare and slows guideline adoption [9,22,45]. Many promising interventions therefore remain in preclinical silos, awaiting coordinated multicenter validation with agreed designs and endpoints [24,101]. A systems-level, interdisciplinary effort is needed to harmonize definitions of injury, align outcome metrics, and integrate precision tools into clinical workflows [36,76].

Important mechanistic uncertainties remain. Even with machine perfusion, we do not fully understand how perfusion pressure, shear, and pulsatility shape microvascular function, or how accumulated metabolites such as succinate influence injury during prolonged runs [8]. Addressing these questions will require studies that couple physiology with unbiased readouts and link them to clinically meaningful outcomes.

Ultimately, success will depend on balancing mechanistic insight with clinical feasibility. This will require continued innovation in perfusate composition and delivery platforms [38,82] and, more broadly, a shift in mindset: preservation as a programmable therapeutic interface rather than a static protocol. With harmonized evaluation and stronger mechanistic grounding, preservation strategies can move from simply maintaining viability to actively rehabilitating and individualizing graft function, bringing precision-driven transplantation closer to routine practice [30,44].

## Figures and Tables

**Figure 2 ijms-26-09515-f002:**
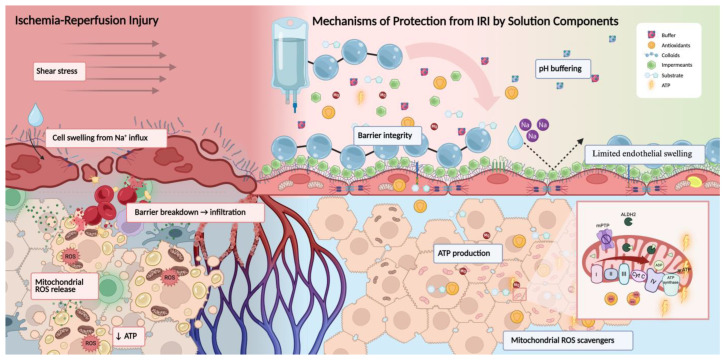
Mechanisms of protection from IRI through preservation solutions. The left panel illustrates key features of IRI during cold ischemia and reperfusion: endothelial glycocalyx degradation, ROS accumulation, decreased ATP, mitochondrial swelling, and parenchymal cell injury. The right panel represents protective effects conferred by preservation solutions, including colloids (PEG-35) that stabilize the endothelial surface and limit shear-induced glycocalyx shedding; impermeants (e.g., lactobionate, raffinose) that prevent osmotic swelling (dashed arrow demonstrates lack of edema) by maintaining extracellular tonicity; buffers (e.g., histidine, phosphate) that stabilize pH; antioxidants that reduce lipid peroxidation; and magnesium, which prevents calcium overload and supports ATP generation. Inset: Within the mitochondria, ALDH2 detoxifies lipid-derived aldehydes, while magnesium supports ATP production and reduces mPTP opening during reperfusion.

**Figure 3 ijms-26-09515-f003:**
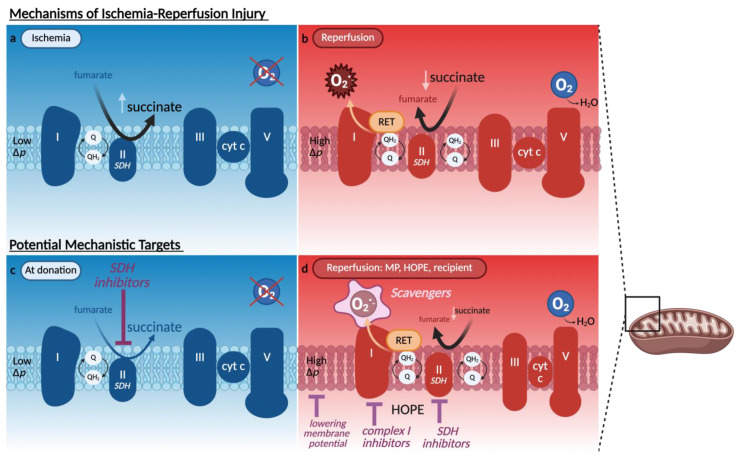
Mitochondrial mechanisms of ischemia–reperfusion injury and therapeutic targets. Panels (**a**,**b**) illustrate the mitochondrial electron transport chain under ischemic and reperfusion conditions, respectively. (**a**) During ischemia, limited oxygen availability halts electron flow, and reverse activity of succinate dehydrogenase (SDH, complex II) leads to the accumulation of succinate, driven by fumarate reduction. (**b**) Upon reperfusion, rapid oxidation of succinate by SDH drives reverse electron transfer (RET) at complex I, generating a burst of ROS at the flavin mononucleotide (FMN) site. This is facilitated by a high mitochondrial membrane potential (Δp). Panels (**c**,**d**) depict mechanistic interventions to reduce injury. (**c**) At the time of organ donation, theoretical SDH inhibition would limit ischemic succinate accumulation. (**d**) During reperfusion, particularly under controlled conditions such as machine perfusion (MP) or hypothermic oxygenated perfusion (HOPE), injury can be mitigated by gradually reintroducing oxygen, lowering Δp, inhibiting complex I, or scavenging ROS. These strategies aim to limit RET-mediated ROS production and preserve mitochondrial integrity. HOPE as a technique slows complex I and II activity. Q/QH_2_ = ubiquinone/ubiquinol; cyt c = cytochrome c; RET = reverse electron transfer; SDH = succinate dehydrogenase; Δp = mitochondrial membrane potential.

**Table 1 ijms-26-09515-t001:** Endothelial-specific effects of preservation solution additives across organ models. SCS = static cold storage; EC = endothelial cell; ICAM-1 = intercellular adhesion molecule-1; ERG = ETS-related gene; arrows depict increase or decrease in component.

Additive	Mechanism of Action	Model & Context	EC-Specific Effects	References
**PEG-35**	Oncotic agent, stabilizes membrane and glycocalyx	IGL-1; Liver and Kidney SCS/HMP	Preserves endothelial ultrastructure and glycocalyx integrity	[2,8]
**Quercetin**	Antioxidant, anti-inflammatory (NF-κB, ICAM-1 inhibition), ERG upregulation	Kidney SCS	↓ ICAM-1, ↑ ERG, ↓ platelet adhesion	[55]
**NO donors**	Vasodilation, anti-inflammatory, ICAM-1/eNOS modulation	Hypothermic machine perfusion (HMP)	↓ ICAM-1, stabilized sinusoidal perfusion	[32,78]

**Table 2 ijms-26-09515-t002:** (**A**,**B**) summarize currently used preservation solutions for each transplantable organ, divided by static cold storage (SCS) and machine perfusion (MP) platforms. For each organ, solutions are listed alongside key physiological vulnerabilities that guide their use, representative literature, and context-specific notes. (**A**) covers SCS strategies, highlighting cold storage solutions routinely used in clinical transplantation (e.g., UW, HTK, Perfadex). (**B**) covers MP strategies, including hypothermic (HOPE) and normothermic (NMP, EVLP, SNMP) protocols. Only solutions with established or emerging clinical use are listed. Experimental approaches (e.g., IGL-2, intestinal perfusion) are excluded unless otherwise stated.

**A. Static Cold Storage (SCS): Solutions Mapped Across Organs**								
**Organ**	**UW**	**HTK**	**IGL-1**	**IGL-2**	**Celsior**	**Custodiol-N**	**Perfadex**	**Key Considerations**
Liver	✓	✓	✓	✓	✓			High metabolic rate, RET-sensitive; bile duct injury risk [19,32]
Kidney	✓	✓	✓					Mitochondrial IRI, edema sensitivity [3]
Heart		✓			✓	✓		High O_2_ demand, edema, redox stress [88]
Lung							✓	Barrier disruption, edema; Perfadex standard [66]
Pancreas	✓	✓	✓					Islet preservation, acidosis risk [89,90]
Intestine	✓		✓					Barrier & mucosal integrity critical [91]
Uterus	✓	✓	✓					Cold ischemia impairs contractility [84,85]
**B. Machine Perfusion (MP): Solutions Mapped Across Organs**								
**Organ**	**Belzer MPS**	**Custodiol-MP**	**Steen**	**RBC-Based**	**Key Considerations**			
Liver	✓			✓	HOPE slows RET; NMP for bile viability [92]			
Kidney	✓	✓		✓	FMN-guided HOPE; DGF reduction [3]			
Heart		✓		✓	Contractility + lactate clearance [7]			
Lung			✓	✓	EVLP for microbe clearance, function [93]			
Pancreas	✓			✓	Islet yield & acinar injury studies [94]			
Intestine	✓ (exp.)			✓ (exp.)	Lactate trend & mucosal barrier [95,96]			
Uterus	✓ (exp.)			✓ (exp.)	Vasoreactivity under investigation [58]			

## Data Availability

No new data were created or analyzed in this study.

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
