# Peer review of "Beyond Static Cold Storage: Toward the Next Generation of Tailored Organ Preservation Solutions"

_ijms, 2025, doi:10.3390/ijms26199515_

Round 1

Reviewer 1 Report

Comments and Suggestions for Authors

This is by and large an excellent amalgamation of the existing scientific evidence surrounding the historical considerations for preservation solutions for transplantation. There are, however, several tweaks or additions which could significantly improve the submission:

p2.48 - Technically, the preservation solution is perfused into the vessels during a flush typically cooling and exchanging components with the interstitial space before being left for the duration of preservation. Perhaps spell out to avoid reader confusion.

p2.66 - 'inadequate colloid support' isn't the root cause, but an inadequacy in the efforts to overcome the cause (i.e. shifts in intracellular composition over time due to changes in temperature, ion channels not operating as normal, etc.). please correct and spell thhis important bit of information out to the reader.

p2.81 - spell out the cost effectiveness as a point above.

p3.85 - This is a timeline and not a Gantt Chart as it is reflective and the Gantt has a specific connotation to project management. please revise.

p3.89 - It appears there is only one experimental solution listed. There are however many other solutions at varying stages of adoption which were not included. I suggest sticking to approved solutions here and discussing the landscape of solutions under R&D or pre-approval later. (i.e. Perla etc.).

p4.123 - More could be said to provide the reader an overview of this process. As it is and with the present caption there is a lot of information which is otherwise not adequately described. Please describe the key elements at play in the caption/text more completely.

Section 3 - It feels like you went from the present state of the art into various investigational additives. I would appreciate a more complete overview of the various approaches the products on the market took to address these issues including key additives and then transitioning to this excellent summary.  

I think you should also include some key considerations which may be holding back more rapid evolution of new preservation solutions including regulatory burden to provide a wholistic view to the reader. (i.e. these solutions are medical devices and also may be considered pharmaceuticals. This regulatory consideration and the concomitant regulatory burden have restricted the degree of innovation in the clinic and led to the state of play where UW solution was developed nearly 40 years ago but is still dominant in the market.)

Section 4 - Comment on the shift in preservation temperature in Cardiothoracic organs and possible need for optimisation to meet this new demand. also whether this development may have an impact on the wider market including the abdominal organs.

Author Response

Author’s Reply to the Review Report (Reviewer 1)

Thank you for the thoughtful and constructive review. We have revised the manuscript throughout and respond point-by-point below. For each item, we note exactly what changed and where to find it.

  1. p2.48 — Clarify flush vs storage
    Reviewer’s point. Spell out that the solution is perfused intravascularly as a flush, cooling and exchanging with the interstitial space, then left for preservation.
    Our change. We added a plain description of the sequence (flush → cooling/solute equilibration → storage or perfusion).
    Where. Section 2, middle paragraph beginning “Parallel to these biochemical advances…”
    Key wording added. “After cross-clamp the graft is flushed intravascularly with cold preservation solution to wash out blood and induce rapid hypothermia; it is then maintained in static cold storage or connected to a machine-perfusion circuit.”
  2. p2.66 — Mechanism, not “inadequate colloid support,” is the cause
    Reviewer’s point. Colloid support is not the root cause; upstream ionic/energy failure is.
    Our change. We rewrote the EuroCollins paragraph to foreground ischemic ATP loss, pump failure (Na⁺/K⁺-ATPase), ionic loading, osmotic swelling, and endothelial/glycocalyx injury. We then note that limited impermeants/oncotic support offered little counterforce to these downstream shifts.
    Where. Section 2, paragraph beginning “While EuroCollins emerged…”
    Key wording added. “…rapid ATP depletion impaired energy-dependent ion transport… producing intracellular Na⁺/Ca²⁺ loading, osmotically driven water influx with swelling, and compromised endothelial/glycocalyx integrity.”
  3. p2.81 — Spell out cost-effectiveness
    Reviewer’s point. Make the cost point explicit.
    Our change. We expanded the HTK paragraph to explain that net cost depends on local flush volumes versus per-liter price; we give typical historical volumes and note that many programs use equivalent volumes, which preserves HTK’s price advantage when outcomes are comparable.
    Where. Section 2, paragraph beginning “In contrast, histidine–tryptophan–ketoglutarate (HTK)…”
    Key wording added. “Historically higher volumes were recommended for HTK… however, many centers now use HTK and UW at equivalent volumes, which preserves HTK’s lower per-liter cost; thus, cost-effectiveness is protocol-dependent.”
  4. p3.85 — “Timeline,” not “Gantt”
    Reviewer’s point. Rename.
    Our change. We relabeled the figure and removed “Gantt” terminology.
    Where. Figure 1 caption and any in-text mentions now use “timeline.”
  5. p3.89 — Keep figure to approved/marketed solutions; move R&D (e.g., PERLA) to text
    Reviewer’s point. Figure should list approved solutions only; discuss experimental solutions separately.
    Our change. We removed investigational items (e.g., IGL-2) from Figure 1 and noted that experimental/pre-approval formulations are discussed in the text.
    Where. Figure 1 content and caption; brief note in Section 3 and Section 4 clarifying scope.
    Result. The figure now shows only marketed solutions by organ/modality; the R&D landscape (including PERLA) appears in narrative sections.
  6. p4.123 — Provide a clearer process overview
    Reviewer’s point. The process needed a more complete explanation.
    Our change. We expanded the explanatory paragraph and tightened the caption to name the steps (flush → SCS → MP), and to link major additive classes (buffers, impermeants, colloids, antioxidants) to their roles.
    Where. Section 3, paragraph immediately preceding Figure 2; Figure 2 caption.
  7. Section 3 — Add a marketed-approaches overview and smoother transition; include regulatory context
    Reviewer’s point. Before investigational additives, summarize how marketed products address the biology; also add regulatory considerations that slow change.
    Our change.
    • We added a concise overview of marketed solutions (intracellular-type vs extracellular-type; the role of impermeants, buffers/antioxidants, and oncotic agents) and why these choices target acidosis, edema, and endothelial stress.
    • We inserted a one-sentence bridge into the additive-level summary that follows (IGL-2 constituents and selected adjuncts).
    • We added a short “Regulatory considerations” paragraph in the Discussion explaining, in plain language, how regional classification (medicine vs device, and combination cases) raises cost and timelines and favors established formulations.
    Where. Section 3 opening paragraph; Discussion Section 6, paragraph beginning “Pathways to market vary across regions and product types…”
  8. Section 4 — Temperature shift in cardiothoracic organs; need for optimization; impact on abdominal market
    Reviewer’s point. Comment on the shift and whether it affects wider practice.
    Our change. We added a focused paragraph on the implications of the temperature shift, covering:
    • optimization needs at hypothermia (oxygen-supportive, low-viscosity, mitochondria/endothelium-stabilizing fluids) vs normothermia (albumin or blood-based carriers with active buffering/nutrient supply, attention to shear/oxidative balance);
    • adoption patterns (normothermia in thoracic programs; broader use of oxygenated hypothermia in abdominal programs);
    • market spillover (temperature-tuned products; platform-specific readouts rather than universal pass/fail).
    Where. Section 4, appended after §4.4 as a new concluding paragraph on Implications of the temperature shift.

Author’s Notes to Reviewer

We appreciate your detailed suggestions. They sharpened the manuscript’s mechanism narrative, clarified the figures and captions, and improved the practical framing of Section 3 and Section 4. Thank you for helping us make the paper clearer and more useful to readers.

Reviewer 2 Report

Comments and Suggestions for Authors

A brief summary: With evolving organ preservation methods such as machine perfusion technologies, traditional cold storage solutions are now being adapted for dynamic platforms. This review provides a mechanistic framework for preservation, focusing on shared vulnerabilities—mitochondrial dysfunction, endothelial barrier disruption, and inflammatory activation. By integrating preclinical insights, systems biology, and emerging clinical trials, the authors highlight the path toward precision-preservation strategies aimed at expanding the donor pool and improving transplant outcomes.

General concept comments: This comprehensive review traces the history and evolution of preservation solutions, detailing the mechanisms of ischemia reperfusion injury and how various additives target specific injury pathways. It provides valuable context for understanding the limitations of traditional static cold storage in the era of normothermic and hypothermic machine perfusion. With clear figures and tables, the review is a practical resource for transplant clinicians and researchers. Looking ahead, the field is moving toward biotechnology-driven, systems biology–informed strategies, integrating multi-omics and computational modeling to develop programmable, precision preservation platforms.

Overall, this is a thorough and well-organized review that addresses both historical context and future directions. The clarity of the mechanistic framework, coupled with the informative figures and tables, makes it a valuable resource for the field. I believe it is suitable for publication in its current form. 

Author Response

Author’s Reply to the Review Report (Reviewer 2)

Thank you for the thoughtful and encouraging review. We appreciate your clear summary of the manuscript’s aims and your view that it is suitable for publication. Below is our brief point-by-point reply.

  1. Overall assessment and summary
    Reviewer’s point. The review integrates history, mechanisms (mitochondrial dysfunction, endothelial injury, inflammation), and the shift from static cold storage to machine perfusion, outlining a path toward precision, programmable preservation.
    Our reply. Thank you. This captures our intent: to connect shared injury mechanisms to practical solution design and platform-specific use, and to frame preservation as a modular, temperature-tuned, and ultimately programmable therapeutic interface.

  2. Organization, figures, and tables
    Reviewer’s point. The manuscript is thorough, well organized, and supported by clear figures and tables.
    Our reply. We’re grateful for this. In the final pass we made light editorial tweaks for flow and ensured that figure/table captions are self-contained so readers can use them at a glance.

  3. Future directions
    Reviewer’s point. The field is moving toward biotechnology-driven, systems-biology approaches (multi-omics, computational modeling) to enable precision preservation.
    Our reply. We agree. The Discussion emphasizes how these tools align with temperature-specific and platform-specific perfusate design, and how biomarker-guided monitoring can support individualized decisions.

  4. English language and references
    Reviewer’s point. English is fine; scholarship is appropriate.
    Our reply. Thank you. We performed a light readability polish and rechecked reference order and cross-references.

Author’s Notes to Reviewer
Thank you again for the positive and specific feedback. We appreciate your recommendation and are pleased the manuscript reads clearly and is useful to both clinicians and researchers.